# Fasting Normoglycemia after Intravenous Thrombolysis Predicts Favorable Long-Term Outcome in Non-Diabetic Patients with Acute Ischemic Stroke

**DOI:** 10.3390/jcm10143005

**Published:** 2021-07-06

**Authors:** Marcin Wnuk, Justyna Derbisz, Leszek Drabik, Maciej Malecki, Agnieszka Slowik

**Affiliations:** 1Department of Neurology, Jagiellonian University Medical College, 30-688 Krakow, Poland; justyna.derbisz@gmail.com (J.D.); agnieszka.slowik@uj.edu.pl (A.S.); 2The University Hospital in Krakow, 30-688 Krakow, Poland; maciej.malecki@uj.edu.pl; 3Department of Pharmacology, Jagiellonian University Medical College, 31-531 Krakow, Poland; leszek.drabik@uj.edu.pl; 4John Paul II Hospital, 31-202 Krakow, Poland; 5Department of Metabolic Diseases, Jagiellonian University Medical College, 30-688 Krakow, Poland

**Keywords:** stroke, thrombolysis, fasting hyperglycemia, fasting normoglycemia, long-term outcome

## Abstract

Background: Only a few studies evaluated the role of fasting glucose levels after intravenous thrombolysis (IVT) in patients with acute ischemic stroke (AIS). Importantly, formal analysis concerning the prognostic role of fasting glucose levels in these patients with and without diabetes mellitus (DM) was not performed. Therefore, we assessed whether fasting normoglycemia (FNG) next morning after AIS treated with IVT was associated with 90-day functional outcome in diabetic and non-diabetic patients. Methods: We retrospectively analyzed 362 AIS patients treated with IVT at The University Hospital in Krakow. FNG was defined as glucose below 5.5 mmol/L. A favorable outcome was defined as modified Rankin score (mRS) of 0–2 at day 90 after AIS onset. Results: At 3-month follow-up, FNG was associated with favorable outcome (87.5% vs. 60.8%, *p* < 0.001) and decreased risk of death (3.1% vs. 18.1%, *p* = 0.002). Independent predictors of a favorable outcome for the whole group were: younger age (HR 0.92, 95%CI 0.89–0.95), lower NIHSS score after IVT (HR 0.70, 95%CI 0.65–0.76), lower maximal systolic blood pressure within 24 h after IVT (HR 0.92, 95%CI 0.89–0.95) and FNG (HR 4.12, 95%CI 1.38–12.35). Association between FNG and mortality was found in univariable (HR 1.47, 95%CI 0.04–0.62) but not in multivariable analysis (HR 0.23, 95%CI 0.03–1.81). In subgroup analyses, FNG was an independent predictor of favorable outcome (HR 5.96, 95%CI 1.42–25.1) only in patients without DM. Conclusions: FNG next morning after IVT is an independent protective factor for a favorable long-term outcome in non-diabetic AIS patients.

## 1. Introduction

The prognostic significance of admission glucose levels in patients with acute ischemic stroke (AIS) treated with intravenous thrombolysis (IVT) is well established [1]. Hyperglycemia on admission has been associated with worse functional outcomes and increased mortality within 3 months after IVT in patients with or without diabetes mellitus (DM) [2].

However, only a few studies so far have evaluated the role of fasting glucose levels after IVT in patients with AIS [3,4]. Fasting hyperglycemia the next day or 2–5 days after IVT was associated with worse 3-month functional outcomes and increased mortality [3,4]. Importantly, the association with outcome was stronger for fasting glucose levels than for admission ones [3]. Diabetes mellitus was not found to be correlated with a 90-day poor clinical outcome as assessed with modified Rankin scale (mRS); however, the formal analysis concerning the prognostic role of fasting glucose levels in AIS patients treated with IVT according to the presence of DM was not performed [3,4]. To the best of our knowledge, no studies so far evaluated the impact of fasting glucose next morning after IVT in diabetic and non-diabetic patients on a 90-day clinical outcome.

The recent Stroke Hyperglycemia Insulin Network Effort (SHINE) trial, performed in AIS patients with concomitant hyperglycemia, evaluated the impact of intensive versus standard insulin therapy during the first 72 h from symptom onset on a 90-day functional outcome [5]. Although the analysis, adjusted for IVT or mechanical thrombectomy (MT) use, did not show any significant difference in long-term prognosis, it occurred that DM was present in around 80% of patients in both treated subgroups, and, consequently, the results may not be generalizable for a whole stroke population, including patients without pre-existent DM [5].

Therefore, the aim of the present study was to search whether fasting normoglycemia (FNG) the next day after IVT was associated with long-term outcomes in a large cohort of AIS patients according to the presence of DM.

## 2. Materials and Methods

The data supporting the results of this study are available from the corresponding author upon reasonable request from any qualified investigator.

### 2.1. Patients

The study was designed as a retrospective analysis of the prospectively collected data of 1209 AIS patients from the Krakow Stroke Data Bank, the registry conducted in the single stroke center, The University Hospital in Krakow, from the year 2007. Finally, the study included 362 AIS patients (29.9%), all of Caucasian origin, treated with IVT between June 2014 and December 2018. We collected the data on demographics, the presence of vascular risk factors, etiology of AIS and National Institutes of Health Stroke Scale (NIHSS) on admission and after IVT. The diagnosis of DM was made as described previously [6]. In brief, patients were diagnosed with DM based on the previous medical history or the use of insulin or antidiabetic oral drugs before the onset of stroke [6].

The outcome was measured with mRS at day 90 from AIS onset, and a favorable outcome was defined as an mRS score of 0–2, similarly to the previous studies investigating the prognostic role of fasting hyperglycemia [3,4]. Additionally, an excellent outcome was defined as an mRS score of 0–1 at day 90 after AIS onset. We also evaluated 3-month mortality. Bleeding brain complications due to IVT were defined in accordance with the ECASS-1 classification [7]. As higher systolic blood pressure (SBP) was found to increase the risk of symptomatic intracranial hemorrhage in the previous stroke registries [8,9], we additionally noted the maximal SBP value within 24 h after IVT.

### 2.2. Glucose Measurements

We evaluated serum glucose levels in each patient the next morning after IVT and overnight fasting. Fasting normoglycemia and hyperglycemia were defined as the glucose levels below 5.5 mmol/L (100 mg%) and equal or above this value, respectively, in accordance with the American Diabetes Association’s Standards of Care [10]. Patients with hyperglycemia greater than 10 mmol/L were treated with four subcutaneous insulin injections daily with doses adjusted to the current level of glucose [11].

The study was approved by the Jagiellonian University Ethical Committee (KBET 54/B/2007). All patients gave informed consent to participate in the study, which was either written or verbal in the presence of at least two physicians in case of inability to use the dominant hand because of AIS.

### 2.3. Statistics

The continuous variables were presented as mean and standard deviation (SD), and in the case of categorical data, counts and percentages were shown. Continuous variables were tested for normality with the use of the Shapiro–Wilk test and then compared, as appropriate, by a Student’s t-test or by the Mann–Whitney U test. The multivariable logistic regression model comprised only those variables that showed a *p*-value of <0.1 in the univariable analysis. We considered a *p*-value of 0.05 (two-sided) as statistically significant and performed all statistical analyses with the use of STATISTICA version 13 (Statsoft Inc, Tulsa, OK, USA).

## 3. Results

### 3.1. Patient Characteristics

The characteristics of 362 AIS patients treated with IVT were summarized in Table 1. Among patients, 108 (29.8%) underwent additionally MT.

Patients with FNG were younger, had lower Body Mass Index (BMI), lower NIHSS score before and after treatment with IVT and lower maximal SBP within 24 h after IVT in comparison to those with fasting hyperglycemia (Table 1).

### 3.2. Association between FNG and Favorable Outcome

#### 3.2.1. All Patients

A favorable outcome applied to 231 (63.8%) patients at 3-month follow-up. Patients with favorable outcome were younger (median, interquartile range IQR 70 (59–79) vs. 78 (69–83) years, *p* < 0.001), less often women (43.7% vs. 59.5%, *p* = 0.005), less often suffered from hypertension (77.5% vs. 95.0%, *p* < 0.001), had lower fasting glucose levels (median, IQR 6.4 (5.5–7.9) vs. 6.9 (6.1–8.6) mmol/L, *p* = 0.004), lower NIHSS score on admission (mean ± standard deviation (SD), 9.7 ± 6.0 vs. 16.3 ± 6.0, *p* < 0.001) and after IVT (mean ± SD, 4.5 ± 4.3 vs. 16.0 ± 7.6, *p* < 0.001), lower value of maximal SBP within 24 h after IVT (145 (126–160) vs. 146 (137–165) mmHg, *p* = 0.048) and less often experienced bleeding brain complications (10.0% vs. 41.3%, *p* < 0.001) compared with the remainder (Appendix A, for subgroup of patients treated only with IVT and without MT see also Appendix A).

Patients with FNG had a higher prevalence of a favorable 3-month outcome than those with fasting hyperglycemia (Table 1, Figure 1). The independent predictors of favorable long-term outcome for the whole group were: younger age, lower NIHSS score after IVT, lower maximal SBP and FNG (Table 2). In the subgroup of patients treated only with IVT (without MT), favorable long-term outcome was predicted by age and NIHSS score after IVT. The association between FNG and outcome for these patients was found in the univariable analysis (Appendix A, Appendix A). The optimal cutoff value of glucose for predicting mRS 0–2 was 5.49 mmol/L. Sensitivity and specificity using this cutoff value were 86.7% and 59.2%, respectively.

#### 3.2.2. Subgroup Analyses According to Diabetes

Patients without DM who presented with FNG in comparison to those with fasting hyperglycemia were younger, had lower NIHSS score after IVT, lower fasting glucose levels, lower maximal SBP and more often underwent additionally MT (Table 1). In non-diabetic patients, variables that predicted a favorable 3-month outcome were: younger age, lower NIHSS score after IVT, lower maximal SBP and FNG (Table 2). In the subgroup of non-diabetic patients treated only with IVT, a favorable 3-month outcome was predicted by age, NIHSS score after IVT, maximal SBP and FNG (Appendix A).

Patients with DM and FNG in comparison to those with fasting hyperglycemia had a lower BMI and lower fasting glucose levels (Table 1). The presence of FNG was not an independent predictor of favorable long-term outcome in diabetic patients in contrast to a younger age, lower NIHSS score after IVT and lower creatinine concentration (Table 2). In the subgroup of diabetic patients treated only with IVT, a favorable long-term outcome was predicted by age, BMI and NIHSS score after IVT (Appendix A).

### 3.3. Association between FNG and Excellent Outcome

#### 3.3.1. All Patients

Patients with FNG had a higher prevalence of an excellent 3-month outcome than those with fasting hyperglycemia (Table 1, Figure 1).

An excellent outcome occurred in 206 (56.9%) patients. Patients with excellent outcome were younger (median, IQR, 69.5 (59–78) vs. 77 (68–83) years, *p* < 0.001), less often women (43.7% vs. 56.9%, *p* = 0.015), less often suffered from hypertension (76.2% vs. 93.8%, *p* < 0.001) and atrial fibrillation (25.2% vs. 34.9%, *p* = 0.049), less often underwent MT (24.3% vs. 37.7%, *p* = 0.007), had lower NIHSS score on admission (mean ± SD, 9.1 ± 5.7 vs. 16.0 ± 6.0, *p* < 0.001) and after IVT (mean ± SD, 3.9 ± 3.8 vs. 14.8 ± 7.6, *p* < 0.001), a lower maximal SBP within 24 h after IVT (median, IQR 145 (126–159) vs. 146.5 (136–166) mmHg, *p* = 0.019), lower fasting glucose levels (median, IQR 6.3 (5.5–7.7) vs. 7.0 (6.1–8.8) mmol/L, *p* < 0.001) and less often experienced bleeding brain complications (9.2% vs. 37.0%, *p* < 0.001) compared with the remainder (Appendix A). In the multivariable logistic regression model, the excellent outcome was predicted by lower NIHSS score after IVT and the presence of FNG (Table 2). In the subgroup of patients treated only with IVT and without MT, an excellent 3-month outcome was predicted by age, NIHSS score after IVT and FNG (Appendix A).

#### 3.3.2. Subgroup Analyses According to Diabetes

For non-diabetic patients, independent predictors of excellent outcome were lower age, female sex, lower NIHSS score after IVT and FNG (Table 2). In the subgroup of non-diabetic patients treated only with IVT, an excellent 3-month outcome was predicted by age and NIHSS score after IVT (Appendix A).

An excellent outcome for diabetics was predicted by a lower NIHSS score after IVT and lower creatinine concentration but not by FNG (Table 2). In the subgroup of patients with diabetes treated with IVT and without MT, the independent predictors of excellent outcome were BMI, NIHSS score after IVT and creatinine concentration (Appendix A).

### 3.4. Association between FNG and Mortality

At a 3-month follow-up, 54 (14.9%) patients died. Patients with FNG had a lower risk of death than those with fasting hyperglycemia (Table 1, Figure 1). The association between FNG and mortality was found in the univariable model for the whole group and the subgroup of patients treated only with IVT (Table 3, Appendix A). The factors that independently predicted the risk of death were older age, higher NIHSS score after IVT and hemorrhagic brain complications for the whole group, and age and NIHSS score after IVT in the subgroup of patients treated only with IVT and without MT (Table 3, Appendix A).

## 4. Discussion

Our study is the first to show that the association between FNG and long-term functional outcome after AIS treated with IVT is limited to the patients without pre-existent DM. This association was present even when the group of AIS patients was restricted to those who were treated only with IVT and without MT. Although this observation might be biased by a small sample size of patients with DM, our results were similar to the conclusions coming from our previous study performed on AIS patients who underwent MT [6]. Similarly, in the recent study of more than one thousand Chinese AIS patients, it was found that admission glucose levels independently predicted worse clinical outcomes after IVT only in patients without DM [12]. Non-diabetic patients seemed to be less adjusted to increased glucose levels than diabetics. This observation also came from a large cohort of more than 20,000 patients in whom fasting or random hyperglycemia increased the risk of transfer to an intensive care unit or in-hospital mortality only in non-diabetics [13]. Moreover, the pathophysiological mechanisms underlying the response to hyperglycemia in patients with and without DM might be different as glucose levels affected enlargement of infarct size in non-diabetic patients in contrast to diabetics [14]. Interestingly, in non-diabetic patients, female sex decreased the chance of a long-term excellent outcome. This observation stayed in line with previous research showing that after adjustment for age, women suffered from more severe AIS on admission and had a worse 3-month functional outcome [15].

Our study revealed that FNG after IVT resulted in a 3 or 4-fold increase in the chance of an excellent or favorable long-term outcome, respectively, at 3-month follow-up in AIS patients. Our results stayed in accordance with previous studies performed on smaller patient populations, which showed that fasting hyperglycemia after IVT increased the risk of worse outcome 3 months after AIS onset [3,4]. However, in our study, the effect size of fasting glucose levels was higher than previously reported. Data coming from a greater study that used a more complex parameter termed the stress hyperglycemia ratio but based on fasting glucose and glycated hemoglobin levels also indirectly supported the role of fasting hyper- or normoglycemia as prognostic factors in AIS patients treated with IVT [16]. In our study, fasting glucose levels influenced long-term outcomes, as did other well-known risk factors such as age and NIHSS score [17], both on admission and after IVT [17,18]. Interestingly, the lower maximal SBP within 24 h after IVT also independently affected the 3-month favorable outcome after AIS. Similarly, the Chinese study of 433 patients treated with IVT supported the prognostic role of lower SBP and revealed that maintaining its levels below 159.5 mmHg increased the probability of a favorable 3-month outcome [19].

We found that FNG decreased the chance of death within 3 months after AIS in patients treated with IVT in univariable analysis. In the previous studies, there was either no analysis concerning the association with mortality performed [4] or, similar to our findings, fasting glucose levels did not independently predict the risk of death in the multivariable analysis [3]. One of the possible explanations of the lack of association between FNG and mortality may be a small number of patients who died at the 3-month follow-up. Other factors might also exhibit a more important risk for long-term mortality after IVT, such as age or bleeding brain complications rate, as was shown in the Pomeranian Stroke Register in Poland during the 3-year post-AIS observation period [20].

Patients with normoglycemia were younger, had lower BMI and less often suffered from DM, suggesting that insulin resistance might play a key role in mediating the risk of a long-term unfavorable outcome [21]. On the other hand, in the Japanese trial of 4655 AIS patients, it was found that the association of insulin resistance with outcome was also maintained in non-diabetic and non-obese patients [22]. Finally, neither BMI nor the presence of DM predicted outcome in the multivariable logistic regression model in our study for the whole group of AIS patients treated with IVT.

Our study has important limitations. First, the character of the study was retrospective. Moreover, we did not monitor for a change in glucose levels in the forthcoming days after IVT. We also did not gather the information on whether patients developed DM after hospitalization. Second, the subgroup analyses, especially according to the DM, may be biased by the small sample size. Third, the confounders, such as age, NIHSS score and SBP value, had a significant influence on the outcome of this study. Fourth, the study included only patients of Caucasian origin; therefore, its results may not be generalized to the patients of other ethnicities. Fifth, the results reported here may not reflect a cause-effect relationship.

## 5. Conclusions

In conclusion, although FNG is an infrequent finding in patients with AIS treated with IVT, it increases the chance of a favorable and even an excellent 3-month outcome in non-diabetics. It seems reasonable to undertake future studies to develop the prognostic scales in AIS patients treated with IVT with FNG as one of the important factors.

## Figures and Tables

**Figure 1 jcm-10-03005-f001:**
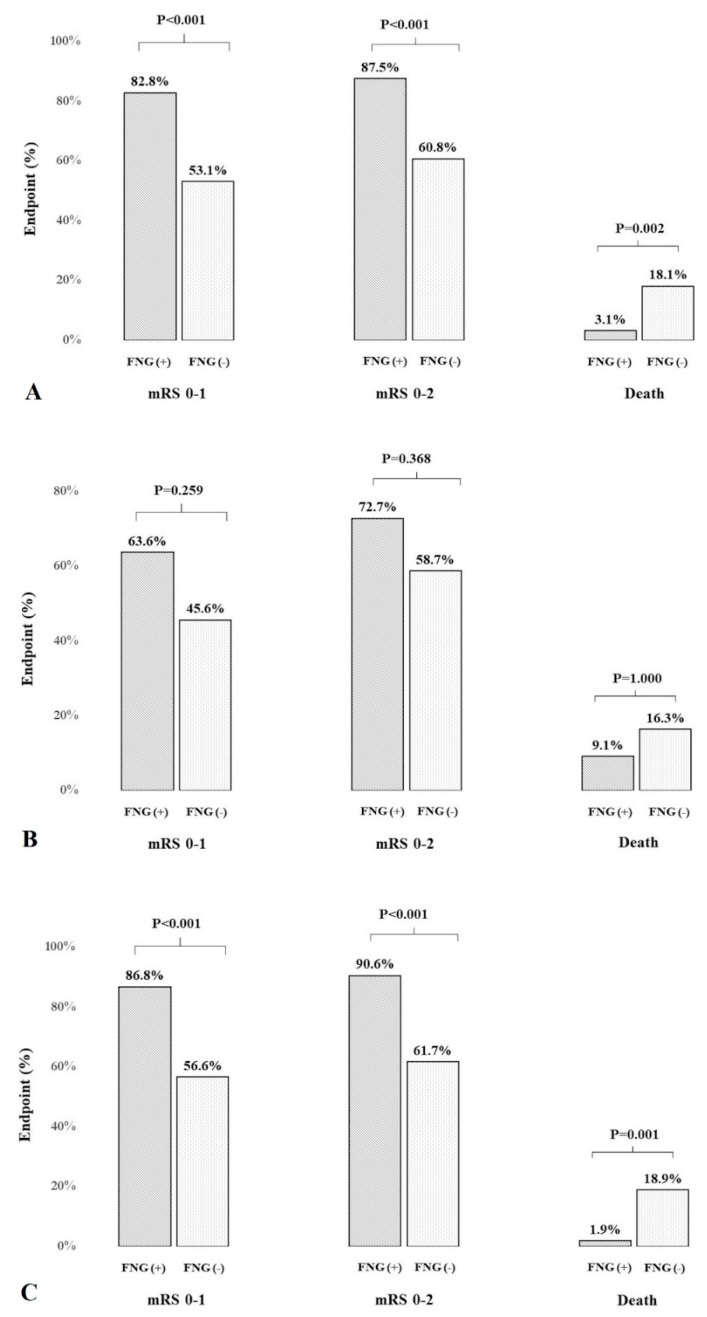
The proportion of patients with a favorable (mRS 0–2) or excellent (mRS 0–1) outcome and those who died (mRS = 6) according to the presence of fasting normoglycemia in the whole group (**A**) and patients with (**B**) and without diabetes mellitus (**C**).

**Table 1 jcm-10-03005-t001:** Baseline characteristics of patients according to glycemia < 5.5 mmol/L and diabetes mellitus (DM).

	Whole Group			With DM			Without DM		
	Glucose < 5.5 mmol/L *n* = 69	Glucose ≥ 5.5 mmol/L *n* = 293	*p*-Value	Glucose < 5.5 mmol/L *n* = 12	Glucose ≥ 5.5 mmol/L *n* = 93	*p*-Value	Glucose < 5.5 mmol/L *n* = 57	Glucose ≥ 5.5 mmol/L *n* = 200	*p*-Value
Age (years)	70 (54–78)	73 (64–82)	0.005	78 (71–80)	74 (65–82)	0.331	65 (53–75)	72 (64–82)	<0.001
Women, *n* (%)	30 (43.5)	147 (50.2)	0.317	4 (33.3)	47 (50.5)	0.360	26 (45.6)	100 (50.0)	0.559
BMI (kg/m^2^)	25.8 (23.4–27.8)	27.1 (24.4–30.1)	0.013	24.9 (23.4–27.0)	29.1 (26.1–31.8)	0.001	25.9 (23.4–28.7)	25.9 (23.9–29.4)	0.578
Hypertension, *n* (%)	54 (78.3)	250 (85.3)	0.150	11 (91.7)	87 (93.5)	0.584	43 (75.4)	163 (81.5)	0.311
Ischemic heart disease, *n* (%)	14 (20.3)	70 (23.9)	0.523	3 (25.0)	30 (32.3)	0.749	11 (19.3)	40 (20.0)	0.907
Atrial fibrillation, *n* (%)	16 (23.2)	90 (30.7)	0.216	3 (25.0)	36 (38.7)	0.528	13 (22.8)	54 (27.0)	0.525
Hypercholesterolemia, *n* (%)	29 (42.0)	88 (30.0)	0.055	7 (58.3)	30 (32.3)	0.108	22 (38.6)	58 (29.0)	0.167
Smoking, *n* (%)	12 (17.7)	43 (15.2)	0.736	1 (8.3)	10 (10.8)	1.00	11 (19.3)	33 (17.2)	0.812
Previous stroke, *n* (%)	12 (17.4)	54 (18.40	0.840	2 (16.7)	21 (22.6)	1.00	10 (17.5)	33 (16.5)	0.852
Stroke etiology, *n* (%) - large-vessel disease - small-vessel disease - cardioembolic - other - undetermined									
	10 (14.5)	39 (13.3)	0.172	4 (33.3)	14 (15.0)	0.534	6 (10.5)	25 (12.5)	0.077
	2 (2.9)	1 (0.3)		0 (0.0)	1 (1.1)		2 (3.5)	0 (0.0)	
	19 (27.5)	100 (34.1)		3 (25.0)	38 (40.9)		16 (28.1)	62 (31.0)	
	5 (7.3)	12 (4.1)		0 (0.0)	2 (2.1)		5 (8.8)	10 (5.0)	
	33 (47.8)	141 (48.1)		5 (41.7)	38 (40.9)		28 (49.1)	103 (51.5)	
Mechanical thrombectomy, *n* (%)	26 (37.7)	82 (28.0)	0.113	1 (8.3)	33 (35.5)	0.978	25 (43.9)	49 (24.5)	0.004
Time from stroke onset to thrombolysis (min)	138 (99–176)	135 (95–183)	0.655	115 (91–156)	135 (96–185)	0.243	140 (100–190)	135 (94–180)	0.998
NIHSS score on admission	10.3 ± 6.6	12.3 ± 6.8	0.026	8.7 ± 4.9	12.3 ± 6.4	0.060	10.7 ± 6.1	12.3 ± 7.0	0.122
NIHSS score after r-tPA	6.3 ± 6.6	9.0 ± 7.9	0.012	7.0 ± 5.1	8.4 ± 7.0	0.665	6.1 ± 6.9	9.3 ± 8.3	0.010
Post-MT hemorrhagic brain complications, *n* (%) - no complication - HI type 1 - HI type 2 - PH type 1 - PH type 2									

	58 (84.1)	229 (78.2)	0.810	11 (91.7)	73 (78.5)	0.718	47 (82.5)	156 (78.0)	0.892
	4 (5.8)	21 (7.2)		1 (8.3)	6 (6.4)		3 (5.3)	15 (7.5)	
	4 (5.8)	20 (6.8)		0 (0.0)	7 (7.5)		4 (7.0)	13 (6.5)	
	2 (2.9)	12 (4.1)		0 (0.0)	4 (4.3)		2 (3.5)	8 (4.0)	
	1 (1.5)	11 (3.8)		0 (0.0)	3 (3.2)		1 (1.7)	8 (4.0)	
Maximal SBP within 24 h after r-tPA (mmHg)	140 (120–150)	147 (135–164)	0.002	140 (125–150)	148 (135–160)	0.285	140 (120–150)	146 (134–165)	0.005
Maximal DBP within 24 h after r-tPA (mmHg)	80 (71–83)	80 (70–90)	0.101	80 (70–80)	80 (70–90)	0.459	80 (72–85)	80 (71–90)	0.124
Fasting glucose (mmol/L)	5.0 (4.7–5.3)	6.9 (6.2–8.6)	<0.001	4.4 (3.6–4.7)	8.1 (6.5–11.5)	<0.001	5.2 (4.9–5.3)	6.7 (6.1–7.8)	<0.001
Creatinine (µmol/L)	78 (69–98)	82 (68–97)	0.572	103 (72–122)	83 (64–99)	0.177	74 (67–95)	82 (69–92)	0.202
mRS 0–1, 90 days, *n* (%)	53 (82.8)	153 (53.1)	<0.001	7 (63.6)	42 (45.6)	0.259	46 (86.8)	111 (56.6)	<0.001
mRS 0–2, 90 days, *n* (%)	56 (87.5)	175 (60.8)	<0.001	8 (72.7)	54 (58.7)	0.368	48 (90.6)	121 (61.7)	<0.001
Death (mRS = 6), 90 days, *n* (%)	2 (3.1)	52 (18.1)	0.002	1 (9.1)	15 (16.3)	1.000	1 (1.9)	37 (18.9)	0.001

Values are presented as *n* (%), mean ± standard deviation, or median and interquartile range. Abbreviations: BMI—body mass index, DBP—diastolic blood pressure, DM—diabetes mellitus, HI—hemorrhagic infarction, mRS—modified Rankin scale, MT—mechanical thrombectomy, NIHSS—National Institutes of Health Stroke Scale, PH—parenchymal hematoma, r-tPA—recombinant tissue plasminogen activator and SBP—systolic blood pressure.

**Table 2 jcm-10-03005-t002:** The multivariable logistic regression model for a favorable (mRS 0–2) and excellent (mRS 0–1) 3-month clinical outcome.

**Favorable Outcome (mRS 0–2)**						
	**Univariable**			**Multivariable**		
**90-day favorable clinical outcome, diabetic + non-diabetic patients**	**HR**	**95% CI**	***p*-Value**	**HR**	**95% CI**	***p*-Value**
Age (per 1 year)	0.95	0.94–0.97	<0.001	0.92	0.89–0.95	<0.001
Sex (female)	0.53	0.34–0.83	0.005	-	-	-
BMI (per 1 unit)	0.96	0.92–1.02	0.163	-	-	-
Atrial fibrillation	0.57	0.35–0.91	0.019	-	-	
NIHSS score after r-tPA (per 1 point)	0.73	0.69–0.78	<0.001	0.70	0.65–0.76	<0.001
Maximal SBP within 24 h after r-tPA (per 1 mmHg)	0.99	0.98–0.99	0.053	0.92	0.89–0.95	0.037
Mechanical thrombectomy	0.67	0.42–1.07	0.091	-	-	-
Hemorrhagic brain complications (ECASS 1–3)	0.16	0.09–0.28	<0.001	-	-	-
Fasting glucose < 5.5 mmol/L	4.52	2.08–9.83	<0.001	4.12	1.38–12.35	0.011
90-day favorable clinical outcome, diabetic patients						
Age (per 1 year)	0.96	0.92–0.99	0.352	0.87	0.80–0.95	0.002
Sex (female)	0.92	0.42–2.03	0.841	-	-	-
BMI (per 1 unit)	0.89	0.81–0.98	0.018	-	-	-
Atrial fibrillation	0.48	0.21–1.08	0.075	-	-	-
Previous stroke	0.41	0.16–1.06	0.067	-	-	-
NIHSS score after r-tPA (per 1 point)	0.77	0.70–0.85	<0.001	0.64	0.54–0.77	<0.001
Hemorrhagic brain complications (ECASS 1–3)	0.25	0.09–0.68	0.007	-	-	-
Creatinine (per 1 µmol/L)	0.98	0.97–0.99	0.019	0.97	0.94–0.99	0.018
Fasting glucose < 5.5 mmol/L	1.88	0.47–1.54	0.375	-	-	-
90-day favorable clinical outcome, non-diabetic patients						
Age (per 1 year)	0.95	0.93–0.98	<0.001	0.94	0.90–0.98	0.01
Sex (female)	0.40	0.23–0.69	0.011	-	-	-
BMI (per 1 unit)	1.02	0.95–1.09	0.585	-	-	-
Hypertension	0.18	0.07–0.47	<0.001	-	-	-
Maximal SBP within 24 h after r-tPA (per 1 mmHg)	0.99	0.98–0.99	0.012	1.03	1.01–1.05	0.007
NIHSS score after r-tPA (per 1 point)	0.71	0.66–0.77	<0.001	0.68	0.62–0.75	<0.001
Hemorrhagic brain complications (ECASS 1–3)	0.13	0.07–0.25	<0.001	-	-	-
Fasting glucose < 5.5 mmol/L	5.95	2.27–15.6	<0.001	5.96	1.42–25.1	0.015
**Excellent Outcome (mRS 0–1)**						
	**Univariable**			**Multivariable**		
**90-day excellent clinical outcome, diabetic + non-diabetic patients**	**HR**	**95% CI**	***p*-Value**	**HR**	**95% CI**	***p*-Value**
Sex (female)	0.59	0.38–0.90	0.015	-	-	-
BMI (per 1 unit)	0.96	0.96–1.01	0.126	-	-	-
Atrial fibrillation	0.63	0.40–0.99	0.049	-	-	-
NIHSS score after r-tPA (per 1 point)	0.73	0.69–0.77	<0.001	0.71	0.66–0.76	<0.001
Maximal SBP within 24 h after r-tPA (per 1 mmHg)	0.99	0.98–0.99	0.014	-	-	-
Mechanical thrombectomy	0.53	0.33–0.84	0.007	-	-	-
Hemorrhagic brain complications (ECASS 1–3)	0.17	0.10–0.30	<0.001	-	-	-
Fasting glucose < 5.5 mmol/L	4.25	2.13–8.47	<0.001	3.47	1.32–9.14	0.012
90-day excellent clinical outcome, diabetic patients						
Age (per 1 year)	0.98	0.94–1.02	0.226	-	-	-
Sex (female)	1.52	0.70–3.32	0.289	-	-	-
BMI (per 1 unit)	0.89	0.81–0.98	0.019	-	-	-
NIHSS score after r-tPA (per 1 point)	0.70	0.62–0.80	<0.001	0.69	0.60–0.79	<0.001
Mechanical thrombectomy	0.42	0.18–1.00	0.050	-	-	-
Hemorrhagic brain complications (ECASS 1–3)	0.27	0.09–0.81	0.019	-	-	-
Creatinine (per 1 µmol/L)	0.99	0.97–0.99	0.040	0.98	0.96–0.99	0.021
Fasting glucose < 5.5 mmol/L	2.08	0.57–7.61	0.267	-	-	-
90-day excellent clinical outcome, non-diabetic patients						
Age (per 1 year)	0.95	0.93–0.97	<0.001	0.96	0.93–0.99	0.02
Sex (female)	0.37	0.22–0.63	<0.001	0.34	0.14–0.87	0.024
BMI (per 1 unit)	1.03	0.96–1.10	0.474	-	-	-
Hypertension	0.21	0.09–0.49	<0.001	-	-	-
Maximal SBP within 24 h after r-tPA (per 1 mmHg)	0.98	0.97–0.99	<0.001	-	-	-
NIHSS score after r-tPA (per 1 point)	0.72	0.67–0.78	<0.001	0.71	0.65–0.77	<0.001
Mechanical thrombectomy	0.59	0.34–1.03	0.065			
Hemorrhagic brain complications (ECASS 1–3)	0.14	0.07–0.28	<0.001	-	-	-
Fasting glucose < 5.5 mmol/L	5.03	2.16–11.7	<0.001	3.47	1.10–12.2	0.035

Abbreviations, see Table 1; ECASS—The European Cooperative Acute Stroke Study.

**Table 3 jcm-10-03005-t003:** The multivariable logistic regression model for the risk of death.

	Univariable			Multivariable		
90-Day Risk of Death, Diabetic + Non-Diabetic Patients	HR	95% CI	*p*-Value	HR	95% CI	*p*-Value
Age (per 1 year)	1.05	1.02–1.08	<0.001	1.07	1.03–1.11	<0.001
Sex (female)	1.48	0.82–2.66	0.189	-	-	-
BMI (per 1 unit)	1.07	0.94–1.08	0.870	-	-	-
NIHSS score after r-tPA (per 1 point)	1.24	1.17–1.31	<0.001	1.22	1.15–1.29	<0.001
Hemorrhagic brain complications (ECASS 1–3)	6.70	3.59–12.5	<0.001	2.66	1.19–5.91	0.017
Maximal SBP within 24 h after r-tPA (per 1 mmHg)	1.01	1.01–1.03	0.014	-	-	-
Hypertension	3.85	1.16–12.78	0.028	-	-	-
Atrial fibrillation	2.02	1.11–3.67	0.021	-	-	-
Platelets (per 105/µL)	0.65	0.40–1.04	0.071	-	-	-
Fasting glucose < 5.5 mmol/L	1.47	0.04–0.62	0.009	0.23	0.03–1.81	0.164
90-day risk of death, diabetic patients						
Age (per 1 year)	1.03	0.98–1.09	0.254	-	-	-
Sex (female)	1.12	0.39–3.26	0.832	-	-	-
BMI (per 1 unit)	1.11	1.00–1.25	0.057	-	-	-
NIHSS score after r-tPA (per 1 point)	1.24	1.12–1.38	<0.001	1.27	1.12–1.42	<0.001
Hemorrhagic brain complications (ECASS 1–3)	4.06	1.30–12.7	0.016	-	-	-
Atrial fibrillation	3.70	1.22–11.2	0.021	5.12	1.31–20.1	0.019
Platelets count (per 105/µL)	0.37	0.12–1.11	0.075	-	-	-
Fasting glucose < 5.5 mmol/L	0.51	0.06–4.31	0.539	-	-	-
90-day risk of death, non-diabetic patients						
Age (per 1 year)	1.06	1.02–1.09	<0.001	1.07	1.02–1.11	0.005
Sex (female)	1.67	0.83–3.38	0.153	-	-	-
BMI (per 1 unit)	0.94	0.85–1.03	0.190	-	-	-
NIHSS score after r-tPA (per 1 point)	1.24	1.16–1.33	<0.001	1.20	1.12–1.29	<0.001
Hemorrhagic brain complications (ECASS 1–3)	8.29	3.92–17.6	<0.001	2.93	1.12–7.69	0.029
Maximal SBP within 24 h after r-tPA (per 1 mmHg)	1.02	1.01–1.04	0.001	-	-	-
Hypertension	5.44	1.27–23.4	0.023	-	-	-
Fasting glucose < 5.5 mmol/L	0.08	0.01–0.62	0.015	0.07	0.01–2.10	0.125

Abbreviations, see Table 1 and Table 2.

## Data Availability

The data supporting the results of this study are available from the corresponding author upon reasonable request from any qualified investigator.

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
