# Peer review of "Fasting Normoglycemia after Intravenous Thrombolysis Predicts Favorable Long-Term Outcome in Non-Diabetic Patients with Acute Ischemic Stroke"

_jcm, 2021, doi:10.3390/jcm10143005_

Round 1

Reviewer 1 Report

  1. Abstract

- Authors should start the abstract with a brief introduction, which should be followed by the aim of the study.

- Line 18 – “according to the presence of diabetes mellitus (DM)” – authors have performed a separate analysis of both diabetic and non-diabetic patients. This sentence does not clarify that. Please consider reframing this sentence.

  1. Introduction

- Recommend citation at the end of the sentence to maintain consistency in the article.

  1. Methods

- Why was the data not included for patients after Dec 2018?

- Line 72 – “The diagnosis of DM was made as described previously.” When was the diagnosis made? Recommend providing more clarification on this.

- What was the method used to assess glucose level, point of care glucose or serum glucose, or both?

  1. Results

- Line 108 – Typically stroke patients undergo mechanical thrombectomy if they meet certain criteria and not due to inefficient IVT. Consider removing “due to inefficient IVT”.

  1. Discussion

- Authors should include confounders such as age, NIHSS, SBP in the limitation section as well. These factors have a strong influence on the outcome of this study.

Author Response

Thank you for your insightful comments and suggestions.

Comments and Suggestions for Authors

1. Abstract

- Authors should start the abstract with a brief introduction, which should be followed by the aim of the study.

Short two-sentence introduction was added at the beginning of the abstract.

- Line 18 – “according to the presence of diabetes mellitus (DM)” – authors have performed a separate analysis of both diabetic and non-diabetic patients. This sentence does not clarify that. Please consider reframing this sentence.

We modified the sentence according to the Reviewer’s comment.

2. Introduction

- Recommend citation at the end of the sentence to maintain consistency in the article.

We modified citations according to the Reviewer’s recommendation.

3. Methods

- Why was the data not included for patients after Dec 2018?

Our stroke database was closed in 2018. Since January 2019 a new pilot program financed by National Health Fund changed the organization of stroke care in Malopolska province with emphasis on mechanical thrombectomy procedures performed in our hospital. Therefore, we constructed a new stroke database since 2019.

- Line 72 – “The diagnosis of DM was made as described previously.” When was the diagnosis made? Recommend providing more clarification on this.

Patients were diagnosed with DM based on the previous medical history or the use of insulin or antidiabetic oral drugs before the onset of stroke. We added this sentence to the Material and Methods section.

- What was the method used to assess glucose level, point of care glucose or serum glucose, or both?

We used serum glucose measurements. This information was added in the Material and Methods section.

4. Results

- Line 108 – Typically stroke patients undergo mechanical thrombectomy if they meet certain criteria and not due to inefficient IVT. Consider removing “due to inefficient IVT”.

We modified the sentence according to the Reviewer’s comment.

5. Discussion

- Authors should include confounders such as age, NIHSS, SBP in the limitation section as well. These factors have a strong influence on the outcome of this study.

We added the sentence in the limitation section that the confounders, such as age, NIHSS score, and SBP value had significant influence on the outcome of this study.

Reviewer 2 Report

In this study the authors replied a their previous research performed in AIS patients treated with MT. In particular, this manuscript was focused on the role of fasting normoglycemia as predictor of outcome in AIS patients receiving alteplase. The authors enrolled 362 patients, but 108 of them were also treated with MT. As the authors well known, patients with LVO have worse outcomes than the remaining AIS subjects; thus, please, exclude patients undergoing MT from the analysis. The authors observed that patients with fasting normoglycemia had a better outcome after IVT. I consider this result not novel. In fact, several previous studies showed that stress-hyperglycemia, a parameter inversely related to fasting normoglycemia, predicted poor outcome in AIS patients receiving alteplase. Moreover, I suggest the authors to include SICH among the outcome measures of their study.

Reviewer 3 Report

Interesting data, well presented. 

  1. line 108 consider deleting "due to inefficient". It is difficult to state the reason for MT was solely due to inefficient IVT. It is recommended of care to receive IVT + MT for eligible LVO patients within the IVT time window.
  2. Table 1. stroke etiology percentage numbers need to be readdressed. Per the table ~50% patients (on both groups) stroke etiology was determined to be secondary to "other" causes. If going by TOAST criteria, this appears to be very high percentage when compared to real world data.    
  3. Consider including all patients were of Caucasian origin as a limitation. 

Author Response

Thank you for your insightful comments and suggestions.

Comments and Suggestions for Authors

Interesting data, well presented. 

  1. line 108 consider deleting "due to inefficient". It is difficult to state the reason for MT was solely due to inefficient IVT. It is recommended of care to receive IVT + MT for eligible LVO patients within the IVT time window.

We modified the sentence according to the Reviewer’s comment.

  1. Table 1. stroke etiology percentage numbers need to be readdressed. Per the table ~50% patients (on both groups) stroke etiology was determined to be secondary to "other" causes. If going by TOAST criteria, this appears to be very high percentage when compared to real world data. 

We corrected the data in the Table 1. It occurred that percentage numbers pertained to stroke of undetermined etiology were wrongly marked in the database as other etiology. The percentage numbers related to other etiology were instead wrongly marked in the database as undetermined etiology. This correction did not affect the results of the study.

  1. Consider including all patients were of Caucasian origin as a limitation. 

We added the sentence in the limitation section that the study included only patients of Caucasian origin, therefore, its results may not be generalized to the patients of other ethnicities.